

# The anti-biofilm effect of silver-nanoparticle-decorated quercetin nanoparticles on a multi-drug resistant *Escherichia coli* strain isolated from a dairy cow with mastitis

Lumin Yu[1,*], Fei Shang[1,*], Xiaolin Chen[1], Jingtian Ni[1], Li Yu[2], Ming Zhang[1], Dongdong Sun[1] and Ting Xue[1]

[1] School of Life Sciences, Anhui Agricultural University, Hefei, Anhui, China
[2] Department of Microbiology and Parasitology, Anhui Key Laboratory of Zoonoses, Anhui Medical University, Hefei, Anhui, China
[*] These authors contributed equally to this work.

Corresponding authors
Dongdong Sun, sunddwj@126.com
Ting Xue, xuet@ahau.edu.cn

## ABSTRACT

**Background.** *Escherichia coli* is an important opportunistic pathogen that could cause inflammation of the udder in dairy cows resulting in reduced milk production and changes in milk composition and quality, and even death of dairy cows. Therefore, mastitis is the main health issue which leads to major economic losses on dairy farms. Antibiotics are routinely used for the treatment of bovine mastitis. The ability to form biofilm increases the antibiotic resistance of *E. coli*. Nanoparticles (NPs), a nanosized, safe, and highly cost-effective antibacterial agent, are potential biomedical tools. Given their antibacterial activities, silver nanoparticles (Ag NPs) have a broad range of applications.

**Methods.** In this study, we performed antibacterial activity assays, biofilm formation assays, scanning electron microscopy (SEM) experiments, and real-time reverse transcription PCR (RT-PCR) experiments to investigate the antibacterial and anti-biofilm effect of quercetin, Ag NPs, and Silver-nanoparticle-decorated quercetin nanoparticles (QA NPs) in *E. coli* strain ECDCM1.

**Results.** In this study, QA NPs, a composite material combining Ag NPs and the plant-derived drug component quercetin, exhibited stronger antibacterial and anti-biofilm properties in a multi-drug resistant *E. coli* strain isolated from a dairy cow with mastitis, compared to Ag NPs and Qe.

**Discussion.** This study provides evidence that QA NPs possess high antibacterial and anti-biofilm activities. They proved to be more effective than Ag NPs and Qe against the biofilm formation of a multi-drug resistant *E. coli* isolated from cows with mastitis. This suggests that QA NPs might be used as a potential antimicrobial agent in the treatment of bovine mastitis caused by *E. coli*.

## INTRODUCTION

Mastitis, an infection or inflammation of the parenchyma of the mammary gland, is considered to be one of the most frequent and troublesome diseases in dairy cows and results in significant economic losses to the dairy industry worldwide (*Halasa et al., 2007*; *Miller et al., 1993*). *Escherichia coli* may cause mastitis which can result in death due to severe endotoxaemia (*Kempf et al., 2016*).

Antimicrobials are used frequently for treatment and prevention of clinical *E. coli* mastitis in cows (*Srinivasan et al., 2007*). However, due to the extensive use of antibacterial agents in bovine mastitis, the emergence of antibiotic-resistant bacteria and the frequent reoccurrence of chronic bovine mastitis has become a critical issue (*Chambers, 2001*; *Gomes & Henriques, 2016*). Another major reason for antibiotic resistance and recurrence of inflammation is bacterial biofilm formation inside the udder tissue (*Atulya et al., 2014*; *Davies, 2003*; *Olson et al., 2002*). Biofilms are defined as communities of microorganisms growing on biotic and abiotic surfaces that tightly adhered to each other in a self-produced extracellular matrix of exopolysaccharide (EPS), proteins, and DNA (*Costerton et al., 1987*; *Prigent-Combaret et al., 2001*). In *E. coli*, various cell surface appendages such as flagella, type I fimbriae, and curli as well as the production of the EPS (such as cellulose, colanic acid, etc.) participate in early adhesion steps (*Cookson, Cooley & Woodward, 2002*; *Pratt & Kolter, 1998*). During the maturation of the biofilm, multiplication of bacterial cells cause an expansion of the microcolonies (*Prigent-Cmbaret et al., 2000*; *Reisner et al., 2003*; *Srinivasan et al., 2007*). Therefore, biofilms can become hundreds of micrometers in depth and display complex structural and functional architecture (*Costerton et al., 1987*; *Pratt & Kolter, 1998*; *Prigent-Cmbaret et al., 2000*; *Prigent-Combaret et al., 2001*). The biofilm can protect *E. coli* against antibiotic therapy and host defense in bovine mastitis (*Dantas et al., 2008*; *Melchior, Vaarkamp & Fink-Gremmels, 2006*; *Xue, Chen & Shang, 2014*). Therefore, the capacity of *E. coli* to form biofilms can result in persistent inflammatory infection in the bovine mammary gland, which is difficult to prevent, control, or eradicate (*Labbate et al., 2004*; *Ren et al., 2005*).

In recent years, nanoparticles, safe, cost effective bactericidal materials, which are gaining popularity, and have been used successfully as carriers of therapeutic agents, in diagnostics of chronic diseases (*Gomes & Henriques, 2016*; *Hong et al., 2008*; *Martinez-Gutierrez et al., 2013*; *Zhang et al., 2008*). In addition, although silver is toxic to microorganisms, it is less dangerous to mammalian cells than other metals (*Morones et al., 2005*; *Sondi & Salopeksondi, 2004*; *Sun et al., 2017*; *Zhao & Stevens Jr , 1998*). Silver nanoparticles (Ag NPs or nanosilver), a kind of nanosized silver particle, are widely used NPs and show strong biocidal effects on a broad spectrum of bacterial pathogens, including *E. coli* (*Morones et al., 2005*; *Sondi & Salopeksondi, 2004*). However, *Choi et al. (2010)* demonstrated that biofilm formation in *E. coli* significantly increased resistance to Ag NPs. Hence, the synergistic effect of Ag NPs and other antimicrobial agents, especially plant-derived drugs, has gotten a lot of attention (*Gomes & Henriques, 2016*). Previous reports show that plant-derived drugs have the advantage of not inducing resistance after prolonged exposure and are easily obtained (*Domadia et al., 2007*; *Ohno et al., 2003*; *Sun et al., 2017*). 3, 3′, 4′,

5, 7-Pentahydroxyflavone (quercetin, Qe), a kind of flavonoid compound, is commonly present in various foods including onions, fruits, and vegetables (*Smith et al., 2016*; *Sun et al., 2017*). Silver-nanoparticle-decorated quercetin nanoparticles (QA NPs) showed highly effective antibacterial activity against drug-resistant *E. coli* (*Sun et al., 2017*). However, whether QA NPs affect biofilm formation of *E. coli* has not been reported. In this study, we examined the effects of QA NPs on growth and biofilm formation of *E. coli* isolated from a dairy cow with mastitis, and explored the regulatory role of QA NPs in biofilm-associated gene expression.

## MATERIALS AND METHODS

### Strains and materials

The multi-drug resistant *E. coli* strain ECDCM1 used in this study was isolated from a dairy cow with mastitis. Strain ECDCM1 was stored at −80 °C. Before each experiment, it was first cultured on Luria-Bertani (LB) agar plates which contained 10 g/L Bacto Tryptone (Oxoid, Basingstoke, UK), 5 g/L yeast extract (Oxoid, Basingstoke, UK), 10 g/L NaCl (Sangon, Shanghai, China), and 20 g/L agar powder (Sangon, Shanghai, China) for 16 h at 37 °C in air supplied with 5% $CO_2$. QA NPs were synthesized according to a previous study (*Sun et al., 2017*).

### Antibacterial activity assays

Antibacterial activity assays were performed according to the following methods (*Chen et al., 2015*). Colonies of strain ECDCM1 were placed into 3 mL of Mueller-Hinton (MH) broth (Oxoid, Basingstoke, UK) and cultivated at 37 °C with shaking at 180 rpm for 16 h. The overnight cultures were inoculated into fresh MH broth and diluted to a final concentration (600 nm) of 0.03 before being dispensed into 96-well plates (Costar, Corning, Steuben, NY) containing Qe, Ag NPs, or QA NPs at the concentrations from 0 mg/mL to 10 mg/ mL, respectively. The bacteria with or without addition of Qe, Ag NPs, and QA NPs were incubated at 37 °C for 20 h, and then 10-fold serial dilutions of cultures were performed by successive transfer (0.1 mL) through six microfuge tubes containing 0.9 mL of MH. Next, 100 μL dilutions were dropped on MH agar plates. After cultivating for 16 h at 37 °C, viable colonies were counted and compared between the control and test groups via their colony-forming units on MH agar plates. The survival rates of the control groups without exposure to Qe, Ag NPs, or QA NPs were designated as 100%. Experiments were repeated three times with three parallels.

### Biofilm formation assay

Biofilm formation in sterile 96-well plates or 15 mm ×150 mm glass tubes was produced as described previously and modified as follows (*Chen et al., 2015*; *Rezaei et al., 2013*). Briefly, *E. coli* strain ECDCM1 was grown for 16 h in LB medium and diluted to a ratio of 1:50 into fresh LB medium containing 0.5% milk solution. The different concentrations of Qe, Ag NPs, and QA NPs, respectively, were added to the LB medium with 0.5% milk solution and *E. coli* cells. The contents were then transferred into 96-well plates and 15 mm ×150 mm glass tubes and incubated without shaking at 37 °C for 72 h and 48 h, respectively. After

incubation, the growth medium was decanted, and the wells were washed three times with sterile phosphate buffered saline (PBS pH 7.4) and air-dried. On observation the biofilms attached to the glass tubes were photographed. In addition, the biofilms attached to 96-well plates were determined quantitatively using crystal violet assay (*Chen et al., 2015*; *O'Toole, 2011*). First, the biofilm-formation cells were fixed with 100% methanol for 5 min. After decanting the methanol, the wells were air-dried and stained for 15 min with 0.1% (w/v) crystal violet (CV, Sangon, Shanghai, China). Next, excess stain was gently rinsed with distilled water and air-dried overnight. Finally, the CV remaining in the wells was dissolved in 33% glacial acetic acid (Sangon, Shanghai, China) and was subsequently read using a MicroELISA autoreader (Thermo Scientific, Pittsburgh, PA) at a wavelength of 492 nm in single wavelength mode. Absorbance data from three replicate wells were averaged to obtain each data point.

## Scanning electron microscopy

Scanning electron microscopy (SEM XL20, Philips, Amsterdam, The Netherlands) was used to investigate the structural modifications of biofilms after treatment with Qe, Ag NPs, or QA NPs. For biofilm formation, the overnight *E. coli* strain ECDCM1 was diluted to a ratio of 1:50 into fresh LB medium complemented with 0.5% milk solution. The cultures were added to 12-well plates (Costar; Corning, Steuben, NY, USA), when necessary, Qe, Ag NPs, or QA NPs were added for a final concentration of 5 $\mu$g/mL. Sterile 18 mm $\times$18 mm coverslips were placed in the wells and served as the attaching surface for the cells. The plates were incubated for 15 h at 37 °C. After incubation, the coverslips were taken out and washed three times with PBS (pH 7.4). The preparation of the samples for SEM was performed as follows(*Sun et al., 2017*) . The samples were fixed overnight with 2.5% glutaraldehyde (Sangon, Shanghai, China) at 4 °C and subsequently dehydrated using serial ethanol concentrations: 30, 50, 70, 80, 95 and 100%. Each ethanol treatment lasted for 20 min at 4 °C and, finally, the cover slips were freeze-dried overnight.

## Total RNA isolation, cDNA generation, and real-time PCR processing

The transcript levels of the biofilm-formation related genes, which are *bcsA* (encoding the catalytic subunit of bacterial cellulose synthase), *csgA* (encoding the major subunit of curli fibers), *fliC* (encoding the basic subunit of *E. coli* flagellar filament structural protein), *fimA* (encoding the major subunit of *E. coli* type 1 fimbriae), *motA* (encoding a protein that enables flagellar motor rotation), and *wcaF* (encoding a putative acetyltransferase involved in colanic acid biosynthesis) were tested by performing real-time reverse transcription-PCR experiments. First, overnight cultures of *E. coli* strain ECDCM1 were diluted to a ratio of 1:50 into fresh LB medium complemented with 0.5% milk solution and 5 $\mu$g/mL of Qe, Ag NPs, or QA NPs were added. The contents were transferred to 12-well plates and grown to the exponential phase at 37 °C without shaking. *E. coli* cells were then collected by centrifugation and resuspended in Tris-EDTA buffer (pH 8.0) containing 10 g/L lysozyme. After incubation for 5 min at 37 °C, total RNA was extracted from cells using the Trizol reagent (Ambion, Austin, TX, USA) and residual DNA was removed using DNaseI (TaKaRa, Dalian, China). Next, real-time RT-PCR was performed using

**Table 1  Oligonucleotide primers used in this study.**

| Primer name | Oligonucleotide (5′–3′) |
| --- | --- |
| rt-16s-f | TTTGAGTTCCCGGCC |
| rt-16s-r | CGGCCGCAAGGTTAA |
| rt-bcsA-f | GATGGTACAAATCTTCCGTC |
| rt-bcsA-r | ATCTTGGAGTTGGTCAGGCT |
| rt-csgA-f | AGCGCTCTGGCAGGTGTTGT |
| rt-csgA-r | GCCACGTTGGGTCAGATCGA |
| rt-fliC-f | CCTGAACAACACCACTACCA |
| rt-fliC-r | TGCTGGATAATCTGCGCTTT |
| rt-motA-f | GGCAATAATGGCAAAGCGAT |
| rt-motA-r | CAGCGAAAACATCCCCATCT |
| rt-wcaF-f | TCTCGGTGCCGAAAGGGTTC |
| rt-wcaF-r | ATTGACGTCATCGCCGACCC |
| rt-fimA-f | TGCTGTCGGTTTTAACATTC |
| rt-fimA-r | ACCAACGTTTGTTGCGCTAC |

the PrimeScript 1st Strand cDNA synthesis kit, SYBR Premix Ex Taq (TaKaRa, Dalian, China), and a StepOne real-time PCR system (Applied Biosystems, Carlsbad, CA, USA). Last, differences of gene expression were calculated by $\Delta\Delta Ct$ (where Ct = cycle threshold) method using the *16S* rDNA gene as the housekeeping gene, normalized by subtracting the Ct value of *16S* cDNA from target cDNA. All of the real-time RT-PCR assays were repeated at least two times with similar results. The primers used in this study are listed in Table 1, and the PCR amplification efficiency was controlled between 1.93 and 2.09.

### Statistical analysis

All data were analyzed using the statistical software SPSS (ver. 19.0, IBM Corp., Armonk, NY, USA) by a one-way ANOVA method; the test results were shown as mean ± SD. The paired *t*-test was used for statistical comparisons between groups. The level of statistical significance was set at a *P*-value of ≤0.05.

## RESULTS

### Antibacterial effect of QA NPs on *E. coli* strain ECDCM1

Since QA NPs were synthesized with the raw material Qe and Ag NPs, the survival rates of *E. coli* strain ECDCM1 exposed to Qe and Ag NPs were firstly detected. As shown in Figs. 1A and 1B, Qe and Ag NPs did not affect the survival rates of *E. coli* strain ECDCM1 at concentrations of 0.5 μg/mL, 1 μg/mL, and 5 μg/mL; and when the concentrations of Qe and Ag NPs reached 10 μg/mL, the survival rates of *E. coli* strain ECDCM1 were approximately 74% and 52%, respectively, compared with the control group without exposure to Qe or Ag NPs. However, except for the 0.5 μg/mL concentration of QA NPs, the survival rate of *E. coli* strain ECDCM1 decreased accordingly with the increase of the concentration of QA NPs, and when the concentration of QA NPs reached 10 μg/mL, the survival rate of cells dropped to 0%. These data indicated that QA NPs had a higher antibacterial activity against *E. coli* strain ECDCM1 than Qe and Ag NPs.

**A**

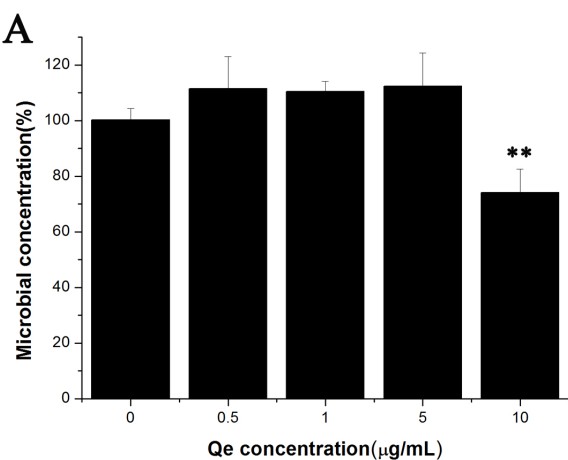

**B**

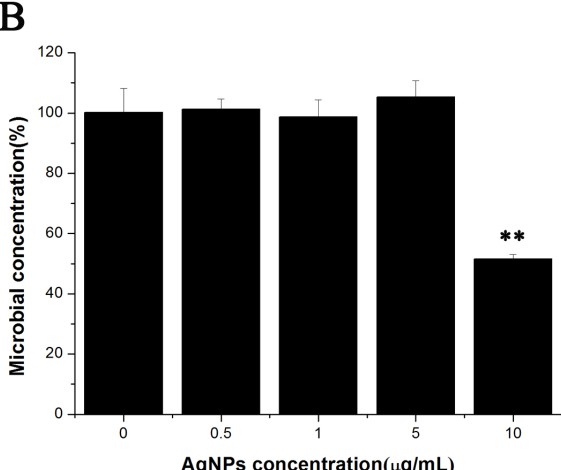

**C**

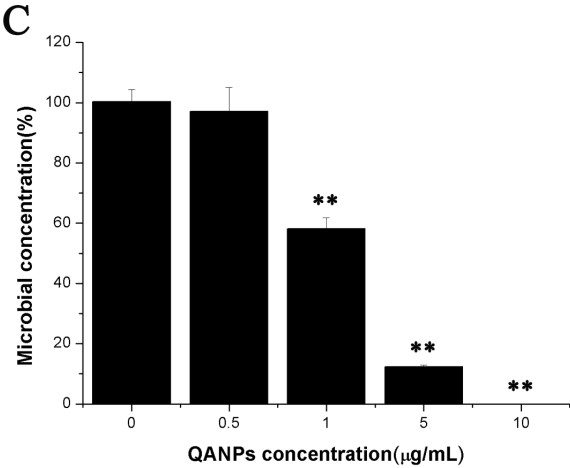

**Figure 1** **Colony-forming unit assays of *E. coli* strain ECDCM1 incubated with Qe, Ag NPs, and QA NPs.** (A) Qe; (B) Ag NPs; (C) QA NPs. The survival rate of the control group without exposure to Qe, Ag NPs, or QA NPs agents was designated 100%. The colony counts of the test group cultured with Qe, Ag NPs, or QA NPs at different concentrations were all compared to those of the control group with no exposure to Qe, Ag NPs, or QA NPs. Error bars indicate standard deviations. Double asterisks (**) represent means significantly different from the control group with no exposure to Qe, Ag NPs, or QA NPs ($P < 0.01$).

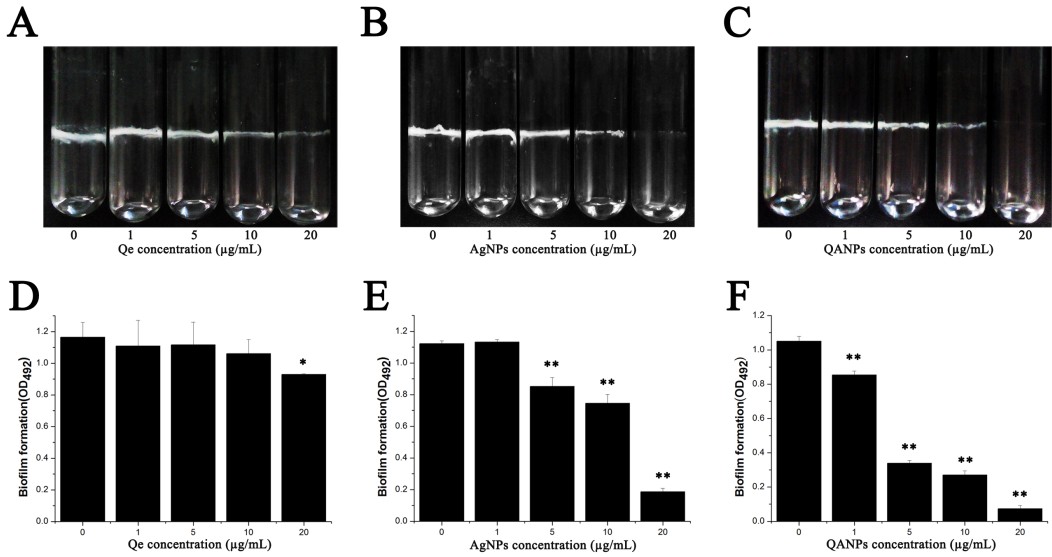

**Figure 2** **Effect of Qe, Ag NPs, and QA NPs on biofilm formation of *E. coli* strain ECDCM1.**
(A–C) photographs of cells adhering to 15 mm ×150 mm glass tubes: (A) Qe; (B) Ag NPs; (C) QA NPs; (D–F) a crystal violet assay and the biofilm quantity being dissolved in 33% glacial acetic acid was measured by optical density at 492 nm: (D) Qe; (E) Ag NPs; (F) QA NPs. Error bars indicate standard deviations. The results represent a mean of three independent experiments. Double asterisks (**) represent means significantly different from the control group with no exposure to Qe, Ag NPs, or QA NPs ($P < 0.01$); an asterisk (*) represents means significantly different from the control group with no exposure to Qe, Ag NPs, or QA NPs ($P < 0.05$).

## Inhibitory effect of QA NPs on biofilm formation of *E. coli* strain ECDCM1

The biofilm was formed both in glass tubes and in a 96-well plate to examine the influence of QA NPs on biofilm formation capacity of *E. coli* strain ECDCM1. As shown in Figs. 2A–2C, Qe, Ag NPs, and QA NPs have no apparent effect on biofilm formation of *E. coli* strain ECDCM1 at concentrations of 1 µg/mL and 5 µg/mL, and at concentration of 10 µg/mL, biofilm formation shows only a slight decrease; however, when the concentration reached 20 µg/mL, biofilm formation is highly inhibited. Especially, at the concentration of 20 µg/mL, QA NPs almost inhibited biofilm formation of *E. coli* strain ECDCM1 on glass tubes to an invisible extent. In addition, the quantity of biofilm formation on the 96-well plates was tested using a MicroELISA autoreader (Thermo Scientific, Pittsburgh, PA). We found that the biomass significantly decreased with the increase of QA NPs compared with that of Ag NPs, and the biomass did not decrease until the concentration of Qe reached 20 µg/mL (Figs. 2D–2F).

To further investigate the effect of QA NPs on the biofilm integrity of *E. coli* strain ECDCM1, we performed a SEM experiment. Results showed that biofilm of the control group were compact and integral (Fig. 3A). When 5 µg/mL concentrations of different agents were added, biofilms were damaged to varying degrees. As shown in Fig. 3B, when the cultures are exposed to Qe, the biofilms are hardly damaged, and the biofilm forms were similar to that of the control group. Meanwhile, biofilms of the Ag NPs treatment group

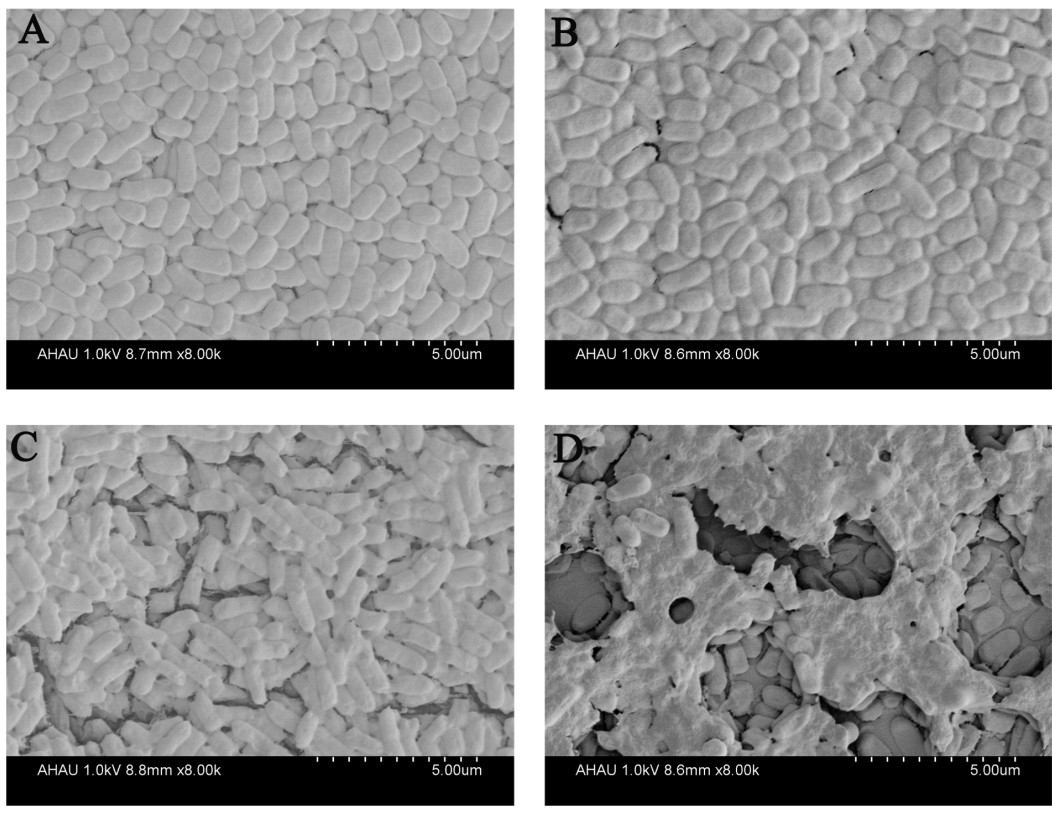

**Figure 3** **Effect of Qe, Ag NPs, and QA NPs on biofilm integrity of *E. coli* strain ECDCM1 were monitored by SEM.** (A) The control group with no antimicrobial agents; (B) *E. coli* strain ECDCM1 with exposure to Qe; (C) *E. coli* strain ECDCM1 with exposure to Ag NPs; (D) *E. coli* strain ECDCM1 with exposure to QA NPs.

were only in slight surface crack (Fig. 3C). However, biofilms of the QA NPs treatment group were seriously damaged (Fig. 3D), indicating that QA NPs highly damaged the biofilm integrity of *E. coli* strain ECDCM1. These results illustrated that QA NPs exhibited stronger inhibition against biofilm formation of *E. coli* strain ECDCM1 than both Qe and Ag NPs.

## QA NPs decrease the transcription of biofilm-associated genes in *E. coli* strain ECDCM1

To further confirm the inhibitory effect of QA NPs on biofilm formation of *E. coli* strain ECDCM1, the transcript levels of biofilm-associated genes (*bcsA*, *csgA*, *fliC*, *fimA*, *motA*, and *wcaF*) were tested by performing real-time RT-PCR experiments. Results showed that the transcript levels of *bcsA*, *csgA*, *fliC*, *fimA*, *motA,* and *wcaF* were significantly decreased upon the addition of Qe, Ag NPs, or QA NPs; moreover, with the addition of QA NPs, the transcript levels of *bcsA*, *csgA*, *fliC*, *fimA*, *motA,* and *wcaF* decreased to lower extents than with that of Qe and Ag NPs (Fig. 4), indicating that the anti-biofilm effect of QA NPs was strongest among the three antimicrobial agents. These results confirmed that

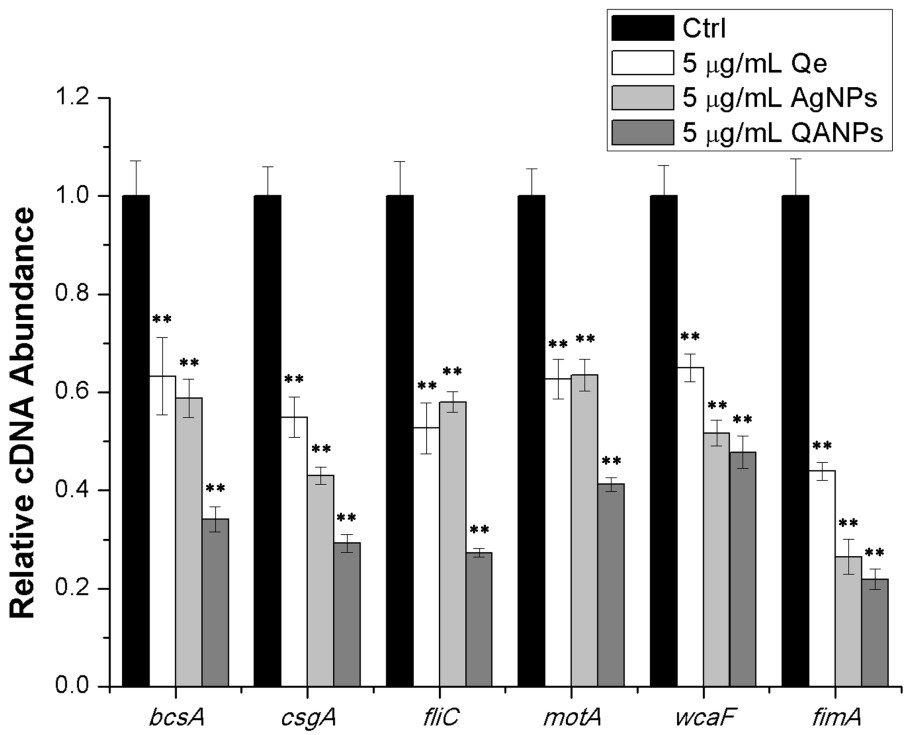

**Figure 4  Comparative measurement of transcription (cDNA abundance) of the biofilm-associated genes in *E. coli* strain ECDCM1.** Relative transcript levels of *bcsA*, *csgA*, *fliC*, *wcaF*, *motA*, and *fimA* were determined by real-time RT-PCR in *E. coli* strain ECDCM1 cultured without or with Qe, Ag NPs, or QA NPs. Error bars indicate standard deviations. Double asterisks (**) represent means significantly different from control group with no antimicrobial agents ($P < 0.01$).

QA NPs efficiently inhibited biofilm formation of *E. coli* strain ECDCM1 by decreasing transcription of biofilm-associated genes *bcsA*, *csgA*, *fliC*, *fimA*, *motA,* and *wcaF*.

## DISCUSSION

Bacteria in biofilms usually induce chronic mastitis characterized by prolonged inflammation and respond poorly to conventional treatments (*Dean, Bishop & Hoek, 2011*; *Secor et al., 2011*). The antibiotic therapy is the important strategy for bovine mastitis treatment (*Saini et al., 2012*; *Saini et al., 2013*; *Yuan, Peng & Gurunathan, 2017*). However, due to the overuse and misuse of antibiotics, the emergence of bacterial resistance among bacteria has become a serious problem in cows with mastitis (*Gomes & Henriques, 2016*). Moreover, the biofilm formation is associated with the increase of bacterial resistance to antibiotics (*Dantas et al., 2008*; *Davies, 2003*). Hence, there is an urgent need for inhibition of the bacterial biofilm formation for controlling bovine mastitis. Plant-derived drugs are new biologically active agents with antimicrobial activity (*Gomes & Henriques, 2016*). For example, the plant extract ursolic acid inhibited biofilm formation of *E. coli* (*Ren et al., 2005*), and the ethanol extract of *Sanguisorba officinalis* strongly inhibited the biofilm formation of methicillin-resistant *Staphylococcus aureus* (MRSA) (*Chen et al., 2015*).

In this study, the strain we used is a clinical multidrug resistant *E. coli* strain which is resistant to a variety of antibiotics, and also has a strong capacity to form biofilms. This kind of strains always causes infections which are difficult to cure. Our data showed that 1 μg/mL QA NPs significantly inhibited the growth of strain ECDCM1. Furthermore, at concentration of 5 μg/mL, QA NPs can disintegrate the structure of biofilm and even kill bacteria in the biofims. These data indicated that QA NPs were comparable to a kind of highly effective antibiotic. In addition, Ag NPs, the most widely used NPs, and Qe, a plant-derived ingredient, are both readily available and not expensive. Thus, the complex QA NPs might have high application value and deveolpment potential in therapy for bovine mastitis.

## CONCLUSIONS

This study found that QA NPs, a composite material combining Ag NPs and the plant-derived drug component quercetin (*Kareem & Al-Hamadani, 2016*), can strongly inhibit the biofilm formation of an *E. coli* strain isolated from a dairy cow with mastitis by decreasing transcription of biofilm-associated genes *bcsA*, *csgA*, *fliC*, *fimA*, *motA,* and *wcaF*.

## ACKNOWLEDGEMENTS

The authors would like to thank Xiuhong Zhou for the technical assistance on scanning electron microscope.

### Funding
This work was supported by the National Natural Science Foundation of China (grants 31672571). The funders had no role in study design, data collection and analysis, decision to publish, or preparation of the manuscript.

### Grant Disclosures
The following grant information was disclosed by the authors:
National Natural Science Foundation of China: 31672571.

### Competing Interests
The authors declare there are no competing interests.

### Author Contributions
- Lumin Yu performed the experiments, analyzed the data, contributed reagents/materials/analysis tools, prepared figures and/or tables, authored or reviewed drafts of the paper, approved the final draft.
- Fei Shang performed the experiments, analyzed the data, contributed reagents/materials/analysis tools, authored or reviewed drafts of the paper.
- Xiaolin Chen and Jingtian Ni analyzed the data, prepared figures and/or tables.

![PeerJ]

- Li Yu and Ming Zhang authored or reviewed drafts of the paper.
- Dongdong Sun conceived and designed the experiments, contributed reagents/materials/analysis tools.
- Ting Xue conceived and designed the experiments, contributed reagents/materials/analysis tools, authored or reviewed drafts of the paper, approved the final draft.

## Data Availability

The raw data are provided in the Supplemental Files.

## Supplemental Information

Supplemental information for this article can be found online at http://dx.doi.org/10.7717/peerj.5711#supplemental-information.

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
