# Peer review of "The anti-biofilm effect of silver-nanoparticle-decorated quercetin nanoparticles on a multi-drug resistant Escherichia coli strain isolated from a dairy cow with mastitis"

_PeerJ, doi:10.7717/peerj.5711_

## Round 0.1 · original submission · Major Revisions

As you can see in the attached reviews, both reviewers have made substantial comments on your manuscript (be sure to notice the extensive PDF document from Reviewer 2). They specifically outline the need for improving the language. Please consider changing your manuscript according to the Reviewers' suggestions, so it can be further considered for publication.

Reviewer 1 ·

Basic reporting

The language of the article needs to be revised and improved.

Experimental design

No comment.

Validity of the findings

No comment.

Additional comments

The manuscript "The anti-biofilm effect of silver nanoparticle-decorated quercetin nanoparticles on a multi-drug resistant Escherichia coli strain isolated from a dairy cow with mastitis" is a continuation of a previous work of the authors and presents interesting findings. However, it must be improved namely in terms of language.
Below, you can find my suggestions and comments.

Abstract: The authors presented two slightly different abstracts. Please rewrite the abstract more succinctly, taking into account the language. Example: Line39/42: repeated information…rephrase.
According to PeerJ journal “Headings in structured abstracts should be bold and followed by a period.” The headings of the abstract are : background, Methods, results and discussion.
Keywords: I suggest adding Quercetin and silver nanoparticles (in substitution of nanoparticles).
Line 109: “3,3′,4′,5,7-Pentahydroxyflavone (quercetin, Qe), a kind of flavonoid compound, can be extracted from the flowers, leaves, and fruits of some plants” give some examples.
Line 128: “…diluted to a final concentration of 0.03 by optical density at 600 nm” change to: diluted to a final optical density (600 nm) of 0.03.
Line 144: “Qe, Ag NPs, or QA NPs were added to the LB medium with 0.5% milk solution and E. coli cells at different concentrations, respectively. Confusing sentence. Rephrase please.
Line 145: Why did you not use the same incubation time?
Line 176: correct the reference (Miller et al. 1993).
Line 209: refer to figure 1C.
Line 230/231: Change to: “Meanwhile, biofilms of the Ag NPs treatment group were only in slight surface crack (3C). However, biofilms of the QA NPs treatment group were seriously damaged (Fig. 3D).
Line 250/253: sentence too long, please rephrase.
Line 266: Reference?
Line 267: “…has been used to treat some infectious diseases” Give some examples please or mention if it refers among other infectious diseases to bovine mastitis.
Line 270/273: repetition of information.
Please review all figures captions (more succinct and objective). They are too long.
Graphic 1C: start the y-axis at 0.

Reviewer 2 ·

Basic reporting

.

Experimental design

.

Validity of the findings

.

Additional comments

pdf attached

Annotated reviews are not available for download in order to protect the identity of reviewers who chose to remain anonymous.

---

## Round 0.2 · accepted · Accept

Thank you for carefully revising your manuscript. You have made the recommended changes and the revised manuscript is clear and understandable. It has been a pleasure working with you and I am looking forward to the publication of your interesting work.

#